# Exploration of clinicians' decision-making regarding transfer of patient care from the emergency department to a medical assessment unit: A qualitative study

Helen Cleak[1☯‡], Sonya R. Osborne[2☯¤‡]*, Julian W. M. de Looze[3,4]

1 School of Allied Health, Human Service and Sport, College of Science, Health & Engineering, La Trobe University, Melbourne, Victoria, Australia, 2 School of Nursing and Midwifery, Faculty of Health, Engineering and Sciences, Centre for Health Research, Institute for Resilient Regions, University of Southern Queensland, Ipswich, Queensland, Australia, 3 Department of Internal Medicine and Aged Care, Royal Brisbane and Women's Hospital, Metro North Hospital and Health Service, Herston, Queensland, Australia, 4 School of Medicine, The University of Queensland, St. Lucia, Queensland, Australia

☯ These authors contributed equally to this work.
¤ Current address: Current addresses: Australian Centre for Health Services Innovation (AusHSI), School of Public Health and Social Work, Queensland University of Technology, Kelvin Grove, Queensland, Australia
‡ These authors are joint senior authors on this work.
* sonya.osborne@usq.edu.au

**Data Availability Statement:** Relevant data are within the manuscript in the form of participant quotes. Raw data transcripts cannot be shared

## Abstract

### Background

Hospitals face immense pressures in balancing patient throughput. Medical assessment units have emerged as a commonplace response to improve the flow of medical patients presenting to the emergency department requiring hospital admission and to ease over-crowding in the emergency department. The aim of this study was to understand factors influencing the decision-making behaviour of key stakeholders involved in the transfer of care of medical patients from one service to the other in a large, tertiary teaching hospital in Queensland, Australia.

### Methods

We used a qualitative approach drawing on data from focus groups with key informant health and professional staff involved in the transfer of care. A theoretically-informed, semi-structured focus group guide was used to facilitate discussion and explore factors impacting on decisions made to transfer care of patients from the emergency department to the medical assessment unit. Thematic analysis was undertaken to look for patterns in the data.

### Results

Two focus groups were conducted with a total of 15 participants. Four main themes were identified: (1) we have a process—we just don't use it; (2) I can do it, but can they; (3) if only we could skype them; and (4) why can't they just go up. Patient flow relies on efficiency in two processes—the transfer of care and the physical re-location of the patient from one

publicly because of the conditions of the ethics approval. Requests in writing for data related to the study titled, Exploration of clinicians' decision-making regarding transfer of patient care from the emergency department to a medical assessment unit: a qualitative study (HREC approval number HREC/16/QPCH/365) can be sent to: The Prince Charles Hospital Human Research Ethics Committee (TPCH HREC EC00168) Attention: Anne Carle, Executive Officer Research, Ethics and Governance Unit, Building 14 The Prince Charles Hospital Rode Road Chermside Qld 4032 Australia Ph: (07) 3139 4500 Research.TPCH@health.qld.gov.au.

**Funding:** This project was funded through a collaborative partnership arrangement between Metro North Hospital and Health Service / Royal Brisbane and Women's Hospital and the Australian Centre for Health Services Innovation (AusHSI) at Queensland University of Technology. SRO and JWMdL applied for the funding to support the study. There is no grant number associated with the funding. Neither the funders nor hospital executive had any role in study design, data collection and analysis, decision to publish, or preparation of the manuscript.

**Competing interests:** The authors have declared that no competing interests exist.

service to the other. The findings suggest that factors other than clinical reasoning are at play in influencing decision-making behaviour.

## Conclusions

Acknowledgement of the interaction within and between professional and health staff (human factors) with the organisational imperatives, policies, and process (system factors) may be critical to improve efficiencies in the service and minimise the introduction of work-arounds that might compromise patient safety.

## Introduction

Hospitals face immense pressures in balancing the number of patients presenting to the emergency department (ED) requiring admission, bed availability, and ED overcrowding [1]. Decision-making to optimise acute care bed utilisation, and in particular in the ED context, occurs in a stressful, time-critical environment and is often exacerbated by target driven pressures to manage patient flows [2, 3]. While the decision to admit and treat individual patients is traditionally viewed as the responsibility of doctors, it is nurses who are largely responsible for managing overall bed capacity and accomplishing a good match between beds requested and beds available [1].

One response to managing patient flow from the ED has been the establishment of Medical Assessment Units (MAUs), which are designed to streamline the capacity of the ED by expediting rapid and comprehensive multidisciplinary assessment of acute medical patients. Evidence suggests that MAUs can provide a more efficient flow of patients accessing hospital services through the ED, reduce ED overcrowding, minimize poor clinical handover, avoid delays, and result in better patient outcomes [3–7]. Nonetheless, there are also suggestions that inappropriate admissions to MAUs are a familiar occurrence and can negatively impact on not only patient care outcomes but also staff workload [8]. A detailed understanding of decision-making processes is required to address the complex risks and issues surrounding the transfer of care between the ED and inpatient medical wards including MAUs [9].

This study is part of larger research program exploring structure and processes surrounding decision-making for streaming patients from the ED to a MAU. This paper presents findings from the inductive qualitative analysis of data collected from a team of medical, nursing, and bed management staff who attended focus groups to explore the decision-making processes used for transfer of care from the ED to the MAU.

## Background

Hospitals are being challenged to meet higher demand for access to their medical and emergency services which can result in the inability of ED patients to be admitted to the hospital in a timely fashion [10, 11]. Research suggests that overcrowding in the ED is mainly the result of a 'systemic lack of capacity throughout health systems', and not necessarily of inappropriate presentations by patients [3]. MAUs have been integrated into health care institutions worldwide, in part to address the increasing numbers of patients attending emergency departments [12–16].

MAUs, similar to short-stay wards or Acute Medicine Units, as they are commonly known in the UK and increasingly in Australia, employ the common principle of providing efficient

and safe care to acutely ill medical patients and includes specific criteria, such as a higher ratio of senior medical staff, established clinical treatment and management protocols, prioritised investigations and urgent treatment coordinated in one clinical area, and leads to patients benefiting from more timely and appropriate clinical care [5, 16–18]. Patients typically experience earlier senior medical involvement [19], reduced length of stay [4, 5, 17–21], reduced risk of unnecessary hospital admissions [18, 19] and investigations [5].

While these tangible benefits are commendable, benefits are dependent upon the effective utilization and organization of MAUs. It is evident from the literature that inappropriate admissions are a familiar issue in many MAUs [8]. McNeill et al. [16] found that patient flow was affected by limited MAU capacity, and inefficiencies in admission processes can lead to ED overcrowding and bed block [22]. The successful operation of MAUs may also be negatively affected by non-MAU patients being admitted as outliers due to a lack of beds elsewhere [18].

Published figures range from a reported rate of 27% of inappropriate admissions in an Emergency Short Stay Unit [23] to less than 50% of the patients intended for admission to a MAU in an Irish health care setting being actually admitted to the unit [24]. In addition, particular populations of patients may be at a greater risk of inappropriate admissions. For instance, in a study of MAUs in Australia, Yong et al. [25] found a high proportion of inappropriate admissions to a medical short-stay unit by elderly patients with comorbidities.

Working in an ED is challenging, with patient flow unpredictable and staff required to make complex, time critical decisions, often with a changing multidisciplinary team and compounded by incomplete and dynamic available information [10, 26]. Thus, inappropriate admissions need to be considered within the wider impact of human factors upon decision-making regarding admission from the ED to the MAU.

The discipline of human factors science is concerned with the understanding of the interaction between people and other elements of a system [27]. According to Health and Safety Executive [28], human factors refers to environmental, organizational and job factors, and human and individual characteristics which influence behaviour at work in a way that can affect health and patient safety. Decisions made regarding one part of the system impact on other parts of the system and how the system behaves as a whole—efficiently or not efficiently. By conceding human limitations and system vulnerabilities, human factors science aims to lessen and mitigate human imperfections to optimise system performance [29]. Consideration of human factors as an inevitable and inherent aspect impacting on decision-making in complex healthcare systems cannot be underestimated.

Decision-making is a complex, multifaceted activity requiring sufficient knowledge and confidence to make the decision. Indeed, it has been highlighted that, in some hospital settings, decision-making regarding bed allocation is largely influenced by administrative staff with or without a medical or nursing background, with the treating medical team not necessarily informed or consulted regarding the decision [24]. It has been argued that a detailed understanding, as opposed to simple solutions, will be required to address the complex hazards and issues surrounding the transfer of care between the ED and medical wards in the hospital [9], including transfers to MAUs.

A two-phased research program was undertaken to determine the effectiveness of clinical criteria for patients admitted from the ED to a MAU within a large quaternary hospital and to explore the influence of potential explanatory factors for deviations in patient streaming disposition. Findings from the first phase, a retrospective medical record review, have been reported elsewhere [30]. The second phase involved shadowed observations of decision-making events and focus groups with decision makers. This paper presents the inductive analysis of qualitative findings from the focus groups conducted in the second phase.

## Methods

### Aim

The aim of this study was to explore, identify, and understand factors influencing the decision-making behaviour of key stakeholders' who are involved in the decision-making process surrounding transfer of medical patients from the ED to the MAU.

### Research questions

The research questions driving this phase of the larger study are:

- What human factors influence key stakeholder's decision-making related to the admission of patients to a MAU?

- What systems factors impede or enable appropriate and timely admission of patients to a MAU?

### Ethics

Low risk ethical approval was obtained from the relevant hospital (approval number HREC/16/QPCH/365) and university (approval number QUT1700000310) human research ethics committees. Following this, site authorization (governance approval) was granted from the study site research governance officer. Data management and storage complied with the National Statement on Ethical Conduct in Human Research [31] and aligned with the relevant university and hospital policies.

### Study design

A qualitative approach using focus groups with health professionals involved in admitting patients from the ED to the MAU was employed to answer the research questions. A semi-structured focus group guide, developed to promote discussion among the participants and keep the conversation on track, was informed by the Theoretical Domains Framework (TDF) [20]. The TDF integrates a number of behaviour change theories into a single, overarching framework and has been used extensively in other health services studies to identify determinants of decision-making behaviour [20, 32].

### Setting

The study was conducted at a large, 929-bed, tertiary and referral hospital in Southeast Queensland, Australia. In 2015, the hospital admitted 100,149 patients, with 74,399 emergency department presentations [33] increasing to 78,000 by November 2019 [34]. The 8-bed MAU was first established and co-located in the ED in 2015; then relocated in 2016 to accommodate an increase to 15-bed capacity. The MAU was designed to improve access and outcomes for patients entering the health system through the ED. The model of care was premised on incorporating front-loaded senior clinician input, enhanced multidisciplinary staffing beyond core business hours, a rapid assessment unit for admissions from the ED, reliable flows out of the unit, and quality clinical handover. Service description, model of care, and MAU principles and processes for admissions, patient flow and transitions of care have been reported elsewhere [30].

### Participants

Participants included a sample of all senior staff involved in making or actioning decisions regarding hospital admission from the Emergency Medical Services (e.g., from the ED) to the

Internal Medicine Service (e.g., to the MAU). We used purposive sampling, adapting a nominated expert sampling recruitment process described by Trotter [35]. Purposive sampling is a type of non-probability sampling widely used in qualitative research for the identification and selection of information-rich cases related to the phenomenon of interest (i.e., decision-making regarding transfer of patient care). The purposive sampling method aligns with the research aim as it resulted in identifying and selecting participants that were especially knowledgeable and were directly involved and experienced in making decisions regarding transfer of patient care from the ED to the MAU) [36], while maximizing effective use of limited resources. Further, the goal of nominated expert sampling was to identify nominated experts with the most extensive expertise in the specific area of social or cultural knowledge [35]. The sample included those eligible staff members who attended at least one of the focus groups.

## Recruitment and consent for focus groups

Potential participants were identified by the Executive Directors of the two services and the sample included medical officers, bed managers, nurse shift coordinators, patient flow administrative officers, and nurse unit managers. Potential participants were invited to participate via an email invitation from the relevant Department Head. All eligible participants were provided with a copy of the relevant Participant Information and Consent Form(s) and informed consent was obtained prior to participation in the focus group. Written consent was obtained from participants prior to the focus groups and confirmed at the start of each focus group prior to participation in the focus group. As part of the informed consent process, potential participants were advised that participation in the focus groups was voluntary and optional and that withdrawal from the study could occur at any time.

## Data collection

Two focus groups were conducted to explore decision-making processes and the approaches, barriers and facilitators to these processes. The focus groups, conducted one after the other, were held during participants' normal working hours, in agreement with the participants and their supervising line managers in order to minimize extra time burden for the participants. The focus groups were attended in a conference room away from participants' usual work areas and facilitated by a trained researcher (HC, a female social worker and academic), with a second trained researcher assisting and taking field notes (SRO, a female registered nurse and academic). Neither HC nor SRO were employed at the study facility or knew the participants in any professional or personal capacity. At the start of each focus group, a third researcher (JdL, a male senior medical officer and clinician researcher) facilitated introductions and then left the room, leaving only participants present. HC and SRO then introduced themselves, their roles in the research, and the purpose of the study at the start of each focus group. Due to the logistical challenges with scheduling, the composition of the groups mainly reflected the two work areas. Internal member checking was facilitated by the second researcher (SRO) who asked for any other opinions (same or different) to each of the prompt questions, if required. Participants in the focus groups include:

- Group 1: Senior ED staff (including medical consultants, senior registrars, nurse unit managers, nurse shift coordinators, senior nurse clinicians)

- Group 2: Senior MAU staff (including medical consultants, senior registrars, nurse unit managers and senior nurse clinicians)

- Patient Flow administrative officers (including bed managers and after hours nurse shift coordinators) were present in both focus groups during a period of crossover.

## Data analysis

Using Braun and Clarke's [37] approach to thematic analysis, the data was analysed in six stages: (1) familiarisation with the data, (2) generating initial codes, (3) searching for themes, (4) reviewing themes, and (5) defining and naming themes; and (6) producing the report.

Focus groups were audio recorded and transcribed verbatim into text files. Two researchers with experience in qualitative research analysed the data using both inductive and deductive approaches. The researchers read through the transcripts first to familiarise themselves with the data. Data was then imported and organised in the NVivo Version 12 [38] software program.

Inductive analysis began by generating initial codes using the constant comparative method as described by Sopcak, Aguilar, O'Brien, et al. [39], drawing on the work of Glaser and Strauss [40] in their development of this methodological framework. In line with constant comparative method, data was analysed in three stages: open coding, axial coding, and selective coding. Data was coded line by line, with each idea given a name (e.g., a word or phrase) that summarised the main idea or concept [open coding) and then codes were collated into categories or potential themes (axial coding). Researchers coded transcripts independently and then met to review, compare and refine codes and search for higher-level themes (selective coding); and then defined and named themes. At each stage, we discussed our respective assumptions and grounds for drawing the conclusions we did. Inconsistencies were resolved through team discussion. We consciously engaged in this process of reflexivity to reduce the likelihood of researcher bias by identifying any personal beliefs that may have inadvertently impacted on the data. Collating codes into themes involved searching for patterns in the data and gathering all data relevant to each potential theme. The final phase was collecting relevant and compelling extract examples and relating these back to the research questions.

The data was then analysed using a deductive approach informed by the Theoretical Domains Framework [20] to identify determinants of decision-making behaviour influencing decision-making with the intention of submitting for publication. Findings from the deductive analysis will be presented elsewhere. The findings that follow are from the first analysis stage, the inductive analysis.

## Findings

Two focus groups were conducted back-to-back with a total of six and nine participants, respectively. The group consisted of senior nursing and medical staff involved in making decisions about the transfer of patients from the point of transfer (ED) to the point of disposition (MAU or home ward); and including professional staff with knowledge of hospital occupancy and responsibility for bed allocation (patient flow department). As suggested by [41], primary factors in determining and assessing sample size sufficiency included the nature of the phenomenon under investigation, the aims and scope of the study and the quality and richness of the data. Although participants represented different staff groups, their responses to the focus groups questions were surprisingly consistent as they repeatedly identified similar issues of concern. Vasileiou, Barnett, Thorpe, et al. [41] suggest that a high level of consensus across the data attests to the richness and volume of data, thus defending the adequacy of the sample size.

Open coding of the original transcripts by two researchers (SRO, HC) resulted in 408 coded extracts across the data from the two focus groups. There was general consensus reported across the two focus groups on the advantages of the MAU in regard to early senior medical intervention and a single point of contact for communication and coordination of activity between the multiple medical specialty teams and diagnostic services inherently involved in the assessment and management of complex patients.

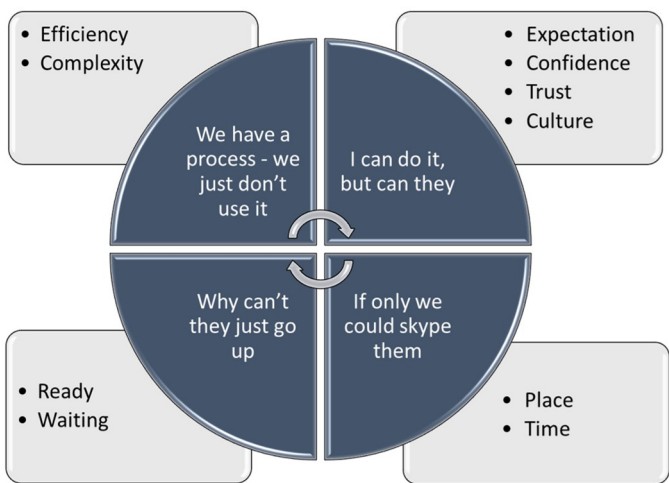

**Fig 1. Thematic depiction of factors related to the process of patient flow from ED to MAU.**

## Key themes identified

Four main themes were identified as we searched for patterns in the data: (1) *we have a process—we just don't use it*; (2) *I can do it, but can they*; (3) *if only we could skype them*; and (4) *why can't they just go up*. Themes and subthemes are presented in Fig 1.

For improved readability, we employed a minimalist approach to smooth some of the verbatim extracts used to illustrate the findings, such as removing filler words and false starts.

## Theme 1—We have a process—We just don't use it

This theme reflects the notion that staff involved in making decisions relevant to patient flow from the ED to the MAU acknowledge or concede that there is a process but not everyone is using the process as it was envisaged when set up.

While all clinicians were positive that the MAU was beneficial, there was less clarity on criteria for patient flow and the process to follow. None of the participants could point out a specific documented hospital policy or procedure that outlined the eligibility criteria for referring patients or the process for transfer of care of patients from the ED to the MAU. All, however, acknowledged there were supposed processes in place but that these were not always being followed. The following exchanges demonstrate not only variation in awareness of protocols to inform or guide decisions and actions in the transfer of care of patients to the MAU but also acknowledgment of tacit hospital processes, such as standing orders, that could be applied to MAU transfer of care decisions.

*Do you think there's any sort of checklist we could come up with for suitability for them [patients] to go up to the ward without them being seen [by the MAU registrar]...*

(*MAU Doctor Registrar*)

*But we already have a process in the hospital, where a patient can go up to the ward unseen, and they've got orders written for four hours, and the [ED] consultant has said, yes, they're stable. I have written medications and fluid orders for four hours, and they go up to [the MAU] . . .that's a written process that we already have–we just don't use it.*

(*ED Nurse Shift Coordinator*)

Two subthemes related to the process were identified throughout the focus groups: the efficiency of the process and the complexity of the process.

**Efficiency.** Despite recognition of unwritten rules for transfer of care of patients from the ED to the MAU, participants discussed the efficiency of the MAU service; owing this to its function as a 'one stop shop' for communication and patient flow management. One of the characteristics of the MAU protocol, as described in the original operational brief, is that there is one single point of contact, a senior MAU clinician, who can accept medical patients on behalf of the various medical service teams. ED doctors described the MAU as an efficient "filter" with the MAU senior registrar acting "like a triage officer" which allowed the ED medical staff to circumvent having to communicate with the multiple subspecialty services under which medical patients might be admitted to hospital.

> . . .theoretically, when you ring the MAU, an SMO [Senior Medical Officer] accept[s] the patient under medicine as a global umbrella, and then they will divvy that up under either the admitting [medical team] for the day or the old team if they're [the patient] is known to a team [from previous admissions]. But, we don't have to worry about that from our [ED doctor] perspective, whereas we used to have to trawl through the chart to see if they [the patient] had been seen in the clinic recently, or something like that.
>
> (*ED Doctor Consultant*)

The ED nurses presented a different perspective on the perceived efficiency and described the scenario of extra unseen work activity and work arounds behind the scenes by Senior Nurses that brought about the perceived or expected efficiency.

> So, these guys [medical officers] now will ask for a bed, and they'll give it to us, the shift coordinators, and they'll just write MAU on it. So, then it's up to us then to do the investigation about old and new [medical team] units. . .and then ask for the bed when we've got that medical unit on the form. . . But it's certainly easier for the docs, because it is just one reg [registrar] to refer. But there's still stuff we have to do in the background to elicit what unit they are, minor but still, just a bit of investigating.
>
> (*ED Nurse Shift Coordinator*)

All participants concurred, to varying degrees, that introduction of the MAU service resulted in a more efficient process compared to before its implementation and acknowledged that there was room for improvement, particularly related to inherent complexity in the process.

**Complexity.** Although the MAU service was perceived to be a more efficient service for transferring care from the ED to the MAU than the previous process, participants talked about the complexity of the process and the many 'hidden' steps involved in the transfer. While acknowledging that the process should be straight forward, participants reported that the process was "complex", "not always clear", and "not as easy as it seemed". There were differing views as to how the process should work and how well the process did work, in terms of getting patients from the ED to the MAU in a timely manner and keeping the safety of the patient at front of mind. One participant reported that transfer of care occurred in two stages: the referral stage and the bed allocation stage.

To complete the referral process, the patient is assessed and referred by a senior ED medical doctor and then the patient is accepted by a senior MAU medical doctor. However, the referral process could take time to complete if this process is not straightforward and other factors

complicated the decision, for example, if the senior MAU medical doctor is not comfortable with the clinical handover given over the phone, prefers to "see" the patient themselves, or requests further investigations. However, once the patient is accepted, bed allocation could be equally as complicated as seen in this description by one participant:

*The doctors would give me a bed request form, and they may write '[MAU] on it, or 'med', that's it. They don't usually write [Ward X], unless the [MAU] reg says to them, oh, this guy was on [Ward X] last week, make him a [Ward X]. Then they'll give me a slip—even then, I'll go back and make sure that's right. Then that bed is requested through bed allocation, so they [bed managers] decide where the patient goes.*

(*ED Nurse Shift Coordinator*)

*But, between ten thirty and seven thirty at night, overnight, bed allocation fax us [shift coordinators] a form where all the empty beds are, and we send them [the patient], first of all, to [MAU]. We try and send them [to MAU], but if there's other wards with empty beds and [MAU] aren't going to decant them we'll go straight to another medical [ward].*

(*ED Nurse Shift Coordinator*)

## Theme 2—I can do it, but can they?

This theme reflects the notion that generally participants were confident in their own knowledge and skills in decision-making and taking actions related to the process of patients moving from ED to MAU; but not necessarily as confident in the system or the capability of other decision makers in the role for which the others had responsibility. Four subthemes related to staff capacity and capability to make decisions regarding patient flow from ED to MAU are role expectations of decision makers, confidence in the system, trust in colleagues to make appropriate decisions, and the culture of the workplace.

**Expectations.** Participants expressed the expectation that individual members of the multidisciplinary team who were being relied upon to make decisions regarding transfer of care from the ED to the MAU would also be expected, rightly or wrongly, to manage the details. The following extract illustrates this point:

*Yeah, I mean, the referral and bed allocation is pretty much in two halves. Referral being medical, bed allocation being nursing . . . we rely on [senior ED nurse] or whoever's doing that job [bed allocation] to sort those details out. . .certainly it's the wrong thing for the senior nurse to do, and it's even more a wrong thing for the senior doctor to do.*

(*ED Doctor Consultant*).

The operational brief describes that the aims of the MAU service are expected to be achieved through early senior clinician intervention, enhanced multidisciplinary staffing and collaborative decision-making centred around the rapid assessment of admissions from the ED. The following extract illustrates the expectations of how the process should always work and the frustration when it does not:

*. . . you've got consultants in the emergency department that are highly trained people that can make these decisions about whether or not a patient is stable to move to a ward, and we tend to ignore that and err on the side of the inpatient team. So, if an emergency*

department consultant is saying that, I believe that this patient is stable enough to be seen on the ward, we get a bed ready, [and] that patient should go. That should be enough.

(Bed Manager).

Participants were all in agreeance that the decision makers should be senior clinicians for a reason and that senior clinician to senior clinician handover was critical to an efficient transfer of care process; as expressed in the following extract:

*One of the big reasons why we do senior referral is so that they're getting a quality referral and —and, um, they can trust what we say, regardless of whether it's me or someone else who's making a referral, . . . or it's a senior person who knows what's going on*

(*ED Doctor Consultant*).

However, sometimes, areas of responsibility are unclear and the nature of working together sometimes more hierarchical than collaborative.

*. . . there could be a perception that that's not particularly interdisciplinary across the, sort of, vertical hierarchy of medicine, but it works very well, I've found. It avoids that, um, decision-making paralysis on behalf of the med reg because you [senior clinician] can, I guess, with more grey hair, synthesise things a little bit more, and make things happen. Whereas, if there are residents [junior clinicians] doing the referrals, um, that can slow things down a bit.*

(*ED Doctor Registrar*)

*From an ED perspective, I think the referral part isn't very collaborative, it's largely medical, they'll do the calling and the talking to the medical assessment unit, . . .and the referral process for that. . .*

(*ED Nurse*)

Despite a consensus on ideal expectations and the positive intentions and acknowledgement of a clear process for the transfer of care of patients from the ED to the MAU, timely patient flow was not always achieved due to a breakdown in the adherence to these processes related to lack of confidence in the system, or trust in the clinical judgement of colleagues.

**Confidence in the system.** The original operational brief developed as the business case for the establishment of the MAU describes a system whereby after a ED Senior Medical Officer (e.g., Senior Registrar or Consultant) makes a decision to admit a medical patient to the hospital, a referral is made to the MAU Senior Medical Officer (e.g., Senior Registrar or Consultant) who then makes the decision to accept the patient. After which, decisions around bed allocation and transfer of patients to the MAU occur in consultation between the ED Senior Nurse (e.g., Shift Coordinator), the MAU Senior Nurse (e.g., Ward Manager), and the Bed Manager. At each of these time points the decision maker needs to have the authority to make the decision that moves the patient along in the process and confidence that the system will work to achieve the desired goal, that is, timely transfer of care from the ED to the MAU. The following extracts illustrate how the lack of confidence in the system and the need to do workarounds can impact the patient flow the process.

*. . .we place too many patients going to MAU at once. It's going to block the emergency department. So, if we can even out some flow by placing some patients to flex beds, other patients to*

*a ward, we'll try and do that just to try and maintain the flow. . .–because if all of the medical patients are going through MAU, we're going to get blocked.*

(*ED Nurse Shift Coordinator*)

*And that's when it comes back to you, . . .actually this system works reasonably well when you're not that busy, but what you really want to know is, if I haven't seen someone, am I confident that they're stable enough that they can wait to be seen before they move.*

(*MAU Doctor Registrar*)

Because the system is made up of actions and interactions between various decision makers across different periods of time, lack of trust in other team members to make timely and appropriate decisions, also impact on the process.

**Trust in others.**   All of the participants were considered senior, experienced clinicians or professionals experienced in their roles with some level of authority of decision-making afforded to them. Participants expressed competence and confidence in their own capabilities in making decisions regarding transferring of care of patients from the ED to the MAU, as illustrated in the following extract:

*I think ED, as a craft group, is probably pretty reasonable at being—identified patients who need a medical admission, and often, sometimes, with orthopaedics or surgery, ah, it may be a little bit unclear, and sometimes they're asking for help, but most of the time if we're asking for an admission, the patient needs an admission.*

(*ED Doctor Consultant*)

However, the group collectively discussed perceptions of an underlying lack of **trust** in 'others' decisions, namely clinical or professional staff who were junior or had not been working at the hospital long enough to understand "the way things worked". This is exemplified in the following extracts.

*Obviously with the turn-over of new registrars, that can take a while for them to get used to that role, but they have to be able to say, 'Yeah, we'll take them [in the MAU].' And then go with that because if they're, pushing back against us [ED]. . .then it sort of defeats the purpose.*

(*ED Doctor Consultant*)

*. . . I guess, you know, if I was a neurosurgical registrar stuck in theatre, and a little baby resident rang me and said I've got a lady with a subdural but she's awake and fine, and that little resident, maybe he was on my rotation last time and he wasn't so hot—I don't know, I mean, it's trust. And I think that, you know, if the [Senior Consultant] rings them up, or [the senior registrar], they may go, yeah, I know that bloke, yeah. . . You know, they've got the trust there —authority and knowledge on that.*

(*ED Nurse Shift Coordinator*)

The discussion continued with participants acknowledging and accepting that trust was needed to enhance efficiency of the process and turned toward conversations about the workplace culture and getting decision-makers to do something different than they were used to.

**Culture.**   While complexity is acknowledged in the decision-making process, the many comments about the failure of the criteria to be applied into practice was often not related to

the content or quality of any guideline or policy, written or unwritten, but to difficulties in changing established behaviours of the clinicians reflecting the culture of the institution. For example, in reference to ED doctors not frequently visiting the "cold zone" of the ED, where patients are relocated following referral for admission and pending transfer to the wards, one participant commented, ". . . it's very difficult to get the ED doctors to come and do anything in cold, but it's still in ED" (MAU Nurse). Another example is about the "need to see the patient". The following extracts depict the frustration with culture interfering with the process:

> *And again, it's this culture of, oh, they can't go anywhere until they've been seen, when they— that shouldn't be the case.*
>
> (*ED Doctor Consultant*)

> *I think it's culture, you know. . . [the boss] puts a very strong front to all the new people coming in saying, here's how it rolls, and if it rolls like this, the machine will work well. Then coming into the clinical space and then seeing a dissonance between how it should 'roll' and how people operate, um, then it's an uncomfortable space to negotiate.*
>
> (*ED Doctor Registrar*)

> *They—these patients come in through the factory of hot [the acute area of the ED]. . . and then they're flipped down to cold. You've got seniority, and go, 'there's a bed'—let's go, let's go, and keep pushing that all the time. Yep, culture's grabbed a lot of people down and [next you're] getting your head ripped off because they [the patient] haven't been seen [by the doctor] or because they have a temperature when they got to the ward. . .all these sort of things.*
>
> (*ED Nurse Shift Coordinator*)

In the end, participants expressed the desire for a more efficient system working as expected, without relying on the way it used to be, and the patients referred to the MAU to just be packaged and transferred to the MAU.

## Theme 3—if only we could skype them

This theme reflects the impressions from participants that there were additional factors outside of their control, that interfered with the smooth disposition of a patient from the ED to the MAU. Three subthemes highlighted the specific pressures of geography or place (distance between the ED and the MAU) and time (time for patient to be accepted to MAU). In addition, culture, or what and how staff were used to the patient flow process happening, was suggested as a barrier for staff embracing the 'new' process.

**Place.** When the MAU was first implemented in the hospital it was co-located in the ED. In order to expand capacity its capacity from 8 beds to 15 beds, the service was shifted to another area in the hospital, away from the ED. This physical distance of the emergency department and MAU was seen as an impediment to the timely transfer of care of patients, particularly from the point of referral.

> *It's a . . .little bit harder because [the MAU] is not that close to emergency. It's a massive environmental thing—It's not like you can just—It's not like you can just, nip in next door.*
>
> (*MAU Nurse*)

The following extracts provide more insight into various views on how location impacted on the patient being accepted by the MAU after being referred by the ED.

*They [MAU Registrar] don't want the nurses to send them upstairs because then they'd [MAU Registrar] have to walk upstairs.*

(*ED Nurse*)

*It's a different area from all the other patients that are waiting to be seen and because it's far away from emergency you don't really have the time just to pop your head in for five minutes and say, okay—that person looks fine.*

(*MAU Doctor Registrar*)

Jokingly, in response to this latter extract, one participant made the following comment, yet all other participants thought it was a good idea:

*I mean, it could be a Go-Pro on there or something like that—then, just Skype the patient in the ward.*

(*ED Nurse Shift Coordinator*)

Apart from distance between the ED and the MAU, the physical distance was directly related to time.

**Time.** Timely patient flow from the ED to the MAU could be impacted not only by a breakdown in adherence to the processes or the tyranny of distance between the two settings but also the actual time required to prepare the patient for physical relocation to the MAU once a referral has been made and accepted. The following extract describes the time it takes to complete necessary activities.

*Part of the complicated flow of moving patients around the hospital is that we'll wait for the MAU reg[istrar], [who] needs to see the patient in the emergency department, reg sees the patient, [then] moving onto his next patient, ah, emergency department—the nurses in cold will then start to package all the paperwork together and that doesn't start to happen until the medical team have seen them, so then the paper work for the patient is packaged, by the nurse in the emergency department, and then we order the transport, and then we wait forty-five minutes for a PSO to come down to the emergency department, pick up the patient, and send them—so all of that packaging can take two hours.*

(*ED Nurse Shift Coordinator*)

Despite the challenges of physical distance and time required to package and transport the patient, in the end the feeling from the group was some frustration in why the medical patients presenting to the ED who deemed appropriate and referred to MAU, could not simply be transferred to the MAU.

## Theme 4—Why can't they just go up?

This final theme, while related to the complexity and efficiency of the process, reflects the dissonance between the patient process and the expectations of how the system and its 'components' should seamlessly work together—and the staff's frustration with that dissonance. Two competing subthemes, 'ready to go' and 'waiting to show', while primarily focused on the

patient, applied directly to the bearing on the process from both the ED and the MAU perspective.

**Ready to go.** Due to a number of factors already described, sometimes patients were **ready to go** from the ED to the MAU; meaning the patient had been assessed by ED medical staff and the decision made by a senior medical clinician to refer the patient to the MAU due to meeting whatever criteria was used to guide the decision. Despite being ready to go up to the MAU patient transfer to MAU was delayed for a number of reasons, such as lack of available beds in MAU or no transport team available. However, participants expressed that more often than not patients were waiting in the ED for the MAU senior registrar to come down to review the patient.

*You look at EDIS [ED patient database] and you look at FOCUS [hospital bed availability database] at four o'clock, they see all these empty beds in the [MAU] and they're ready to go. You look at EDIS—bed ready—bed ready—bed ready—but we can't move the patients because they haven't been seen.*

(Bed Manager)

*That then spills over to the days that [the MAU] has ten empty beds and being they're ready, but everyone's just gone just—oh well, they haven't been [to see the patient].*

(*ED Nurse Shift Coordinator*)

The longer patients' transfer to MAU was delayed, the more likely it was for the patient to be moved to the "cold" zone, a section of the ED for patients awaiting further assessment, diagnostic testing results, treatment, consults, or allocation and transfer to an available bed in the hospital. Participants expressed concern for patients waiting in cold because of a lower nurse-to-patient ratio, the flawed thinking that patients in the cold zone did not require as frequent monitoring, the actual risk of patients deteriorating during the delay in transfer to MAU, and the situation of patients being transferred to MAU in a worse health condition then when they were originally referred. The following extracts illustrate these points:

*And you certainly don't want to be—receiving patients from cold who've been there for, you know, twelve hours. . . If they could come quicker, we could get them settled and sorted quicker, we could get them fed, we could get them [settled]. . .*

(*MAU Nurse*)

*Obviously, they're [patients from ED] coming up to the ward without the criteria, and then we have to start the whole UCR [urgent clinical review] process which calls the [MAU] doctor away from emergency [ED].*

(*MAU Nurse Shift Coordinator*)

**Waiting to show.** Also discussed were situations when MAU staff were waiting for patients to show up to be admitted to the unit from the ED or waiting for patients currently in the MAU to be discharged from the MAU to make room for incoming patients. From the Bed Manager and ED perspective, both situations were seen as "empty beds in [MAU]", whether visibly empty or empty 'on paper', and resulted in patients waiting longer than necessary in the ED. The impact for the bed manager was on the ability to efficiently manage bed

allocations and patient flow. In any case, the waiting to physically re-locate patients to the MAU was a source of frustration.

> *. . . one of the frustrating things is having the empty beds, um, up on [the MAU], and having patients in the emergency department, and they should be going up to the [the MAU] unseen [by the MAU registrar], unless they're clinically unstable. That's the role of [MAU] and that's why it has the registrars that it has, and it's frustrating that doesn't happen.*
>
> (*Bed Manager*)

> *I find a lot of the time after hours when I'm calling down to cold to send the patients up. I'm saying, hey, I'm calling for these cold patients, we haven't seen them yet. They'll say, okay, no worries. But, it's still a while for them to come up.*
>
> (*MAU Nurse Shift Coordinator*)

Many of the participants suggested that patients should "just go up" to the MAU and that the paperwork and investigations could follow. There was discussion around the concept of "push" and "pull", and perhaps assumptions about the patient flow process, as illustrated in the following extracts.

> *I don't know whether there needs to be a bit more of [MAU] pushing to get their patients out [of MAU] so that they can pull [patients] from [ED] and—because then we're calling [MAU] to say, you've still got all these patients on your screen, they're awaiting movement, has any-one called?*
>
> (*ED Nurse Shift Coordinator*)

> *But I think a lot of it is not only is there no pull, there's active push back [from MAU] as opposed to push from ED, because occasionally, presumably, [MAU] does have a bed, and patients are staying in cold for hours.*
>
> (*ED Doctor Consultant*)

> *I'd say we probably don't call, like, I think we just maybe think that [ED] know that we're happy to accept it, but I don't know if you guys [ED] are aware of that. Like, if they're stable, we're always happy to take them if they haven't been seen.*
>
> (*MAU Nurse Shift Coordinator*)

Four main themes and ten subthemes were identified from analysis of patterns in the data. Themes are meant to capture something important in relation to the overall research questions [37]. The research questions driving this study were about the role of human and systems factors on decision-making around the transfer of care of medical patients from the ED to the MAU. The themes identified reflected a somewhat unbalanced combination of both as presented in Fig 2.

## Discussion

This study is the first to offer insights into factors impacting on the decision-making process related to transferring of care of patients from the ED to a MAU—a process which occurs in two distinct and complex stages: patient referral and bed allocation. Participants were able to

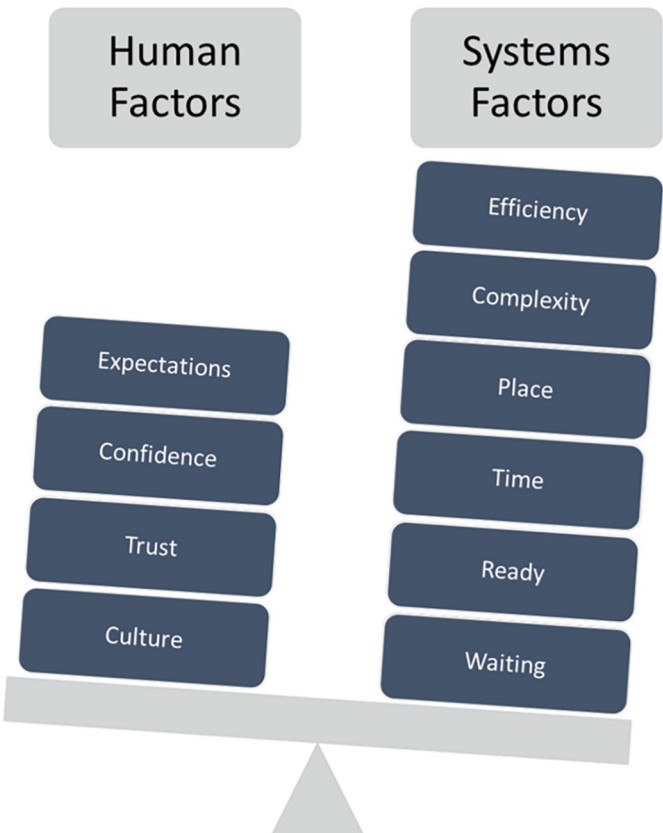

**Fig 2. Human and systems level factors impacting on transfer of care decisions and actions.**

describe their perceptions of the role of the MAU and its advantages, such as a single point of contact when dealing with multiple subspecialties and early medical officer intervention which results in patients getting admitted to the right medical team in a timely and efficient manner. However, the findings also suggested that factors other than clinical reasoning are at play in influencing decision-making behaviour. Despite belonging to different staff work groups, congruence between responses was very evident. Participants were genuinely concerned for the safety of medically complex patients presenting to the ED and viewed decisions to transfer care for these patients to the MAU as the optimal choice for timely comprehensive assessment and management. We triangulated the findings from this study with data collected from our recent audit of transfer of care, decision-making and patient disposition for 712 medical patients presenting to the ED [30]. We found that although almost two-thirds of patients were appropriately transferred to the MAU, a large proportion (16/22, 73%) of patients with contra-indications for admission to the MAU were inappropriately transferred to the MAU (95% CI: 50 to 89%; p-value = 0.05) [30], thus, corroborating the concern for patient safety expressed by participants in the focus groups. The discussions in the focus groups reflected a conscious awareness and reflection about quality improvement and how to make transfer of care more efficient while maintaining patient safety.

The deficits in the implementation of decision-making actions were seen as a risk factor for compromised patient safety by resulting in longer than desired lengths of stay in ED, particularly in the 'cold zone'. The cold zone typically has lower skill mix and staff-patient ratios.

Consequently, timely recognition and response to signs and symptoms of deterioration of patient health status may be compromised; which has implications for patient safety.

Internal patient streaming in the ED using 'zones' based on acuity has been introduced to improve the efficiency of throughput in the ED [42]; however, this may have had a paradoxical effect on the efficiency of transfer of patients to the MAU. Focus group participants in our study expressed concerns for safety of patients waiting in the 'cold' zone to be seen by a senior MAU medical officer and the risk of this 'forgotten' patient deteriorating. Direct transfer of the patient from the ED 'hot' zone after ED referral to the MAU, instead of to the 'cold' zone to await MAU assessment, would eliminate the risk of patient deterioration in the ED and provide the opportunity for timely follow up assessment and diagnostics in the MAU guided by some standing orders.

The emergence of the MAU model of care requires not only trust in the senior staff in ED to make appropriate clinical judgement and complimentary logistical bed allocation decisions that support appropriate and timely patient referral to the MAU but also the willingness and preparedness on the part of the MAU receiving team to take responsibility for the patient based on these decisions [11]. Smooth patient flow through the acute system contributes significantly to the patient experience as well as enabling the efficient use of the limited resources available [43].

## Human factors

Culture is well acknowledged as being a set of shared ideologies, values, assumptions and beliefs that shape human behaviours and their interactions with each other and the workplace, including interactions with organisational systems and processes. Culture drives how and why staff might accept or reject changes to 'the way things are done around here'. Our findings suggest that there is an ethos within the hospital that patient care and throughput processes are being undermined by a culture of mistrust and lack of confidence in colleagues' capability and judgement, and in the system. Fuelling this adverse culture may be the fear of negative consequences, risk aversion and risk management considerations, or lack of specific domain knowledge, particularly that of junior staff. This may have contributed to inconsistent application of the guidelines and subsequent workarounds which resulted in hospital beds being misused or underused, which is a theme reported elsewhere [44, 45].

Workarounds in our study were typically initiated by nurses and took the form of 'matchmaking', a concept described by Allen [1] whereby nurses and bed managers negotiated the needs of the patient with bed availability to optimise hospital bed capacity, particularly out of hours. Workarounds sometimes bypassed the initial recommendations for transfer from the senior medical decision-maker, who often did not have access to previous patient history or bed movements in the hospital obtained from the hospital administrative computer systems. Triangulation with data from our previous study [30] validates this unintended aspect of workarounds; the audit found that admissions from ED after hours, when bed allocation decisions were made by a sole decision-maker (the After Hours Nurse Manager) and pragmatically based on bed capacity and availability, were less likely to be transferred to the MAU (Adjusted odds ratio: 2.09; 95% CI: 1.18 to 3.84; p-value = 0.014). Hospitals evaluating their MAUs may benefit from identifying and exploring the reasons for and the nature of workarounds and consider incorporating these activities into standard MAU processes.

The findings also reflect a culture within the hospital bed management system where the influence of the knowledge and skills of nursing staff was undermined by the culture of medical hegemony and constraining organisational rules inherent in most hospital organisations. Allen [1] reports similar findings that the nurses' knowledge and skills went largely

unexploited and that hospital management needs to ensure that the lines of communication between health disciplines are strengthened. Busby & Gilchrist [46] observed interactions between medical and nursing staff in intensive care and found that although consultants identified that nurses had a high level of knowledge about patients, they asked junior doctor for details about the patient on most (76%) occasions and medical staff asked for the opinion of nurses on only four occasions throughout the nine months of the study. Another study reflected the dominant position of the medical worldview in key decision-making in intensive care [47]. A study of decision-making about post-surgery bed allocation in ICU [48] concluded that by establishing rules for decision making, interprofessional relationships as well as communication between departments could be improved. However, with respect to transfer of care from ED to MAU, although participants in the focus groups acknowledged unwritten rules, there was also a tendency to circumvent these rules based on individual knowledges of systems and processes. Participants recommended parallel processes of patient referral (medical) and bed allocation (nursing and bed managers), as well as better understanding and appreciation of each other's knowledge, skills, capability and scope of practice. This might be a more productive way to utilise the various skill sets of clinicians and professional staff to make appropriate, timely clinical and logistical decisions for referral and bed allocation.

## Systems factors

In this study, systems factors were the major influence on ED to MAU transfer of care and patient flow decision-making. Several authors have reported similar findings in the literature. For instance, poorly defined admission policies and procedures have been identified as a complicating factor for admission and discharge decision-making [19]. The development of rigorous and clearly defined processes of admission and discharge of eligible patients are seen as vital to manage patient flow through to the MAU [49].

It is possible that unanticipated external factors may influence the use and application of admission checklists. For instance, Yong et al. [25] found that external factors such as time and day of admission may also be important in inappropriate admissions [25]. Notwithstanding high-risk immediate emergency care, the use of protocols and checklists have been cited as leading to a loss of professional autonomy [26] and a resultant reluctance of some medical staff to allow complex patients, with low acuity, to be admitted to the MAU for initial assessment and management [18].

An recent Australian study used a model of care where stable patients with more complex problems were seen early by a senior clinician to make decisions on investigation, treatment and disposition [43]. The premise behind this model was that most blocks in patient flow arose from delays in decision-making, which could be improved with senior medical intervention early in the patient journey. The MAU model uses this principle and aimed to take advantage of the advanced decision-making skills of senior clinicians in making early management and disposition decisions. Yet, in our study, practice on the ground was somewhat different. Participants reported that senior medical staff were often occupied with other responsibilities and unavailable in the ED to initiate the referral of patients suitable for admission to the MAU or accept the referral. One strategy to mitigate this deviation from the checklist approach could be to allow referral and acceptance of referral from other senior staff, such as the ED Nursing Shift Coordinator and the MAU Nursing Unit Manager, respectively. This delegation of decision-making responsibility, built into the process, could improve bed utilisation, particularly when the ED Nursing Shift Coordinator has direct access to the hospital systems tracking bed occupancy and patient admission history, and can coordinate the physical transfer of the patient with both the bed manager and the MAU. Implementation of such a process may

be challenging, however, based on previously identified cultural barriers to redefining professional roles in the healthcare environment [1, 47, 48].

A final consideration is the consensus from participants in the focus group was that the physical distance of the MAU, located on another floor in our study site, from the ED interfered with the timely referral and transfer of patients to the MAU. Especially during periods of intense demand on staff, the time the journey took for the referring senior MAU medical clinician to go to the ED to assess a patient and accept the referral was problematic. Hospitals planning to establish a MAU should consider co-location, that is, designing the MAU to be physically adjacent to the ED, to avoid creating a physical barrier to the best use of this service.

## Limitations

The small sample size may be seen as a limit to the generalisability of the findings. As is the case with purposive sampling (a type of non-probability sampling method), we were less concerned with the total population of ED, MAU and bed management staff or the proportion who participated in the focus groups. In comparison, the ability to make accurate generalizations from random sampling (a type of probability sampling method) is highly dependent on sample size. In qualitative research, however, "the sample sizes are typically so small that even random sampling would yield too little accuracy for meaningful generalizations" (p. 725) [50]. The focus group as a data collection method aimed to obtain data from a purposely identified group of key informants (i.e., senior staff involved in making or actioning decisions regarding hospital admission from the ED to the MAU), rather than from a statistically representative sample of a broader population of ED, MAU, and bed management staff. Purposive sampling also allowed us to recruit specific participants because of the various characteristics and perspectives they would bring to the group discussions [36]. For example, capturing data from the point of view of clinicians and professional staff, senior nursing and senior medical staff, and staff from all three areas i.e. MAU, ED and patient flow services, provided the opportunity for enhancing the richness of the data.

The number of focus groups may also be seen as a limitation; however, in their analysis of 40 focus groups, Guest, Namey, and McKenna [51] demonstrated that more than 80% of all themes are discoverable within two to three focus groups. This was confirmed in a study of 10 focus group discussions conducted by Hennink, Kaiser, and Weber [52] to assess saturation. Hennink, et al. [52] concluded that while issues could be identified from one focus group and two focus groups afforded a better understanding of the issues, more focus groups garnered little additional information.

The purpose of this study was to explore factors influencing decision-making related to medical patient flow from the ED to a MAU. The data did not allow us to explore the appropriateness of any decisions or any patient outcomes as a result of decisions made. However, interviewing key stakeholders in addition to the retrospective audit [30] in Phase 1 of the research program strengthened our findings by providing different perspectives from key professional groups (that is, medical doctors, registered nurses, bed allocation professionals) responsible for the transfer of care of patients from the ED to the MAU—from referral to bed allocation. While it could have been viewed as another limitation in having a mixed group of professionals, it became clear during the focus groups that all participants worked together as a team and were comfortable sharing their viewpoints with each other in the open forum.

While the data did not allow us to estimate the frequency with which interruptions to the process or breakdowns in the system occurred during the transfer of care from ED to MAU, which would have been beneficial information for administrators and clinicians, our study was not designed to pursue this line of inquiry.

Finally, the study was conducted in a single-institution and findings might not be generalisable. However, we have provided data on the study hospital for comparison with other institutions which will allow decisions about applicability and transferability of the findings to other institutions.

## Conclusion

Health-care systems face immense challenges to safeguard the quality of patient care in an era of austerity. Some form of a MAU has now become mainstream in acute hospitals since their inception over a decade ago. However, their effectiveness in improving transfer of care and patient flow between the ED and the MAU relies on the presence of senior staff who can make early decisions and have responsive MAU staff who can initiate patients care plans judiciously. We acknowledge that our interpretation of the data is not the only possible interpretation. The findings suggest that attention to both human and systems factors is critical to improve efficiencies in service. Acknowledgement of the interaction within and between professional and health staff (humans) with the organisational imperatives, policies, and process (system) may foster a culture of trust at the individual and systems level within hospitals so that patient care and throughput can be improved.

## Supporting information

**S1 File. Focus group guide.**
(DOCX)

## Acknowledgments

Many thanks to Dr Katie Page for her contributions during the initial discussions about the study and for facilitating the initial stakeholder engagement meeting that preceded this phase of the research program.

## Author Contributions

**Conceptualization:** Sonya R. Osborne, Julian W. M. de Looze.

**Data curation:** Helen Cleak, Sonya R. Osborne.

**Formal analysis:** Helen Cleak, Sonya R. Osborne.

**Funding acquisition:** Sonya R. Osborne, Julian W. M. de Looze.

**Investigation:** Sonya R. Osborne.

**Methodology:** Sonya R. Osborne, Julian W. M. de Looze.

**Project administration:** Sonya R. Osborne.

**Resources:** Sonya R. Osborne, Julian W. M. de Looze.

**Supervision:** Sonya R. Osborne.

**Writing – original draft:** Helen Cleak, Sonya R. Osborne.

**Writing – review & editing:** Helen Cleak, Sonya R. Osborne, Julian W. M. de Looze.

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
