## [Decision Letter · Decision Letter 0]

15 Jun 2021

PONE-D-21-10148

“Why can’t we just skype them!” Exploration of clinicians’ decision-making regarding transfer of patient care from the emergency department to a medical assessment unit: a qualitative study

PLOS ONE

Dear Dr.Sonya Osborne ,

Thank you for submitting your manuscript to PLOS ONE. After careful consideration, we feel that it has merit but does not fully meet PLOS ONE’s publication criteria as it currently stands. Therefore, we invite you to submit a revised version of the manuscript that addresses the points raised during the review process.

ACADEMIC EDITOR:

The paper appears very interesting as it focuses on improving the evaluation process of ED patients for reducing the inappropriateness of hospital admissions. The results may be useful to future scientist conducting research in other contexts.

However, in order for it to be published on PLOSONE, some changes need be made:

Re-evaluation of the first part of the title so that the content of the article is clearerDeepen the discussion showing how this model can be extended to improve the process of assigning admissions in the Medical Area and specifically in the Internal Medicine Departments. Add bibliography on the procedures for admission to the medical area and any similar supporting models.Comprehensive response to the Reviewer's 3 instructions

The decision is justified on PLOS ONE’s publication criteria. 

We look forward to receiving your revised manuscript.

Kind regards,

Filomena Pietrantonio

Academic Editor

PLOS ONE

Journal Requirements:

2. Please include a copy of the interview guide used in the study, in both the original language and English, as Supporting Information, or include a citation if it has been published previously.

Additional Editor Comments (if provided):

The paper appears very interesting as it focuses on improving the evaluation process of ED patients for reducing the inappropriateness of hospital admissions. The results may be useful to future scientist conducting research in other contexts.

However, in order for it to be published on plosone, some changes need to be made:

1. Re-evaluation of the title so that the content of the article is clearer

2. Deepen the discussion showing how this model can be extended to improve the process of assigning admissions in the Medical Area and specifically in the Internal Medicine Departments. Add bibliography on the procedures for admission to the Medical Area and any similar supporting models.

3. Comprehensive response to the Reviewer's 3 instructions

Reviewers' comments:

Reviewer's Responses to Questions

**Comments to the Author**

1. Is the manuscript technically sound, and do the data support the conclusions?

Reviewer #1: Yes

Reviewer #3: Yes

2. Has the statistical analysis been performed appropriately and rigorously? 

Reviewer #1: Yes

Reviewer #3: N/A

3. Have the authors made all data underlying the findings in their manuscript fully available?

Reviewer #1: Yes

Reviewer #3: No

4. Is the manuscript presented in an intelligible fashion and written in standard English?

Reviewer #1: Yes

Reviewer #3: Yes

5. Review Comments to the Author

Reviewer #1: Dear,

I read your work with pleasure and interest. Your ideas are original and in my opinion they should be adopted in any hospital.

I would like to keep in touch with you and your research group.

Best wishes,

Reviewer #3: This is an interesting paper about ED staff dynamics behind the admission of patients to MAU.

The paper is well structured and conceived. The methodology is not particularly strong however, due to the design being a qualitative research conducted on a sample from a single institution. Primary data is unfortunately not available, and there is of course no trace of statistical analysis due to the research being qualitative and not quantitative.

It is therefore hard to generalize the results, although they may be useful to future scientist conducting research in the same field.

The authors did a good work in identifying the main factors implicated in their institution dynamics, presented their findings in an understandable form and acknowledged the limitations within the study itself, wich I believe can be recommended for publication as soon as the following minor points are solved:

A) Participants: it was chosen to have a purposive sample of staff involved in the studied behaviour. Please expand on this concept, explaining:

I) Why this type of sampling was optimal to the research goal, compared to others.

II) If (and why) was any measure of bias reduction (I.E. sample-cherrypicking) taken, if any.

III) What was the proportion of the sample, in relation of the total staff population.

B) The Authors correctly followed the COnsolidated criteria for REporting Qualitative studies (COREQ); however, not all COREQ Items were reported in text. some were N\\A, but two can be easily reported:

No.4) Gender

No. 15) Presence of non-participants

Please provide such information in-text and update Additional File 2 accordingly.

6. PLOS authors have the option to publish the peer review history of their article (what does this mean?). If published, this will include your full peer review and any attached files.

Reviewer #1: **Yes: **francesco rosiello

Reviewer #3: **Yes: **Antonio Vinci

---

## [Author Response · Author response to Decision Letter 0]

23 Jul 2021

RESPONSE TO REVIEWERS

ACADEMIC EDITOR 

Comment 1: The paper appears very interesting as it focuses on improving the evaluation process of ED patients for reducing the inappropriateness of hospital admissions. The results may be useful to future scientist conducting research in other contexts. 

Author Response: We would like to thank the Academic Editor for these positive comments.

Amended Text: N/A

Comment 2: However, in order for it to be published on plosone [sic], some changes need to be made: 

Author Response: We will address the remaining comments point by point and highlight the location of relevant changes in the revised manuscript. 

Amended Text: N/A

Comment 3: Re-evaluation of the first part of the title so that the content of the article is clearer

Author Response: Thank you for this suggestion. We have amended the title to reflect the content of the article more clearly. 

Amended Text: New Title: “Exploration of clinicians’ decision-making regarding transfer of patient care from the emergency department to a medical assessment unit: a qualitative study.”

Comment 4: Deepen the discussion showing how this model can be extended to improve the process of assigning admissions in the Medical Area and specifically in the Internal Medicine Departments. 

Author Response: We thank the Editor for this suggestion and have amended the discussion accordingly. The discussion section has been amended in several places, highlighted for your convenience. 

Location of Changes to Manuscript: Section: Discussion; Page 32: Lines 628-732.

Comment 5: Add bibliography on the procedures for admission to the medical area and any similar supporting models. 

Author Response: Procedures for admission to the medical assessment unit are detailed in The Early Patient Intervention Centre Operational Brief [2016]. This is an internal hospital document of the study site and not publicly available. However, we have added a citation to our published paper which provides more details about the MAU’s service description, model of care, principles and processes. 

Amended text: “The 8-bed MAU was first established and co-located in the ED in 2015; then relocated in 2016 to accommodate an increase to 15-bed capacity. The MAU was designed to improve access and outcomes for patients entering the health system through the ED. The model of care was premised on incorporating front-loaded senior clinician input, enhanced multidisciplinary staffing beyond core business hours, a rapid assessment unit for admissions from the ED, reliable flows out of the unit, and quality clinical handover. Service description, model of care, and MAU principles and processes for admissions, patient flow and transitions of care have been reported elsewhere (30).”

Location of changes to manuscript: Section: Setting; Page 9; Lines 161-169

Comment 6: Comprehensive response to the Reviewer's 3 instructions 

Author Response: This has been attended to. See responses below for Comment Nos. 8-21. 

Amended Text: See changes to manuscript below for Comment Nos. 8-21

REVIEWER #1

Comment 7: I read your work with pleasure and interest. Your ideas are original and in my opinion they should be adopted in any hospital. I would like to keep in touch with you and your research group. Best wishes, 

Author Response: We would like to thank Reviewer #1 for these positive comments and will endeavour to make contact after the review process is completed. 

Amended Text: N/A

REVIEWER #3

Comment 8: This is an interesting paper about ED staff dynamics behind the admission of patients to MAU. 

Author Response: We would like to thank Reviewer #3 for this positive comment. 

Amended Text: N/A

Comment 9: The paper is well structured and conceived. 

Author Response: We would like to thank Reviewer #3 for this positive comment. 

Amended Text: N/A

Comment 10: The methodology is not particularly strong however, due to the design being a qualitative research conducted on a sample from a single institution. 

Author Response: We acknowledge Reviewer #3’s opinion about qualitative research but would like to take the opportunity to educate the reviewer on the purpose and value of qualitative research. All research questions cannot be answered with numbers (quantitative). Qualitative research allows researchers to provide “rich descriptions of complex phenomena; track unique or unexpected events; illuminate the experience and interpretation of events by actors with widely differing stakes and roles; give voice to those whose views are rarely heard; conduct initial explorations to develop theories and to generate and even test hypotheses; and move toward explanations” (Sofaer 1999, p1110). 

Amended Text: No changes made to manuscript.

Comment 11: Primary data is unfortunately not available, and there is of course no trace of statistical analysis due to the research being qualitative and not quantitative. 

Author Response: We agree with Reviewer #3’s observation that there is “no trace of statistical analysis due to the research being qualitative…”. However, we would like to point out that in qualitative research using focus groups or interviews to collect data, the ‘primary data’ is the transcripts of the discussions of what the participants say during the focus group or interview. We have included several rich text descriptions of the primary data to illustrate and justify claims, as is conventional practice when reporting findings of qualitative analysis. 

Amended Text: No changes made to manuscript.

Comment 12: It is therefore hard to generalize the results, although they may be useful to future scientist conducting research in the same field. 

Author Response: The goal of qualitative analysis is not statistical generalizability. Qualitative analysis enables analytical or theoretical generalizability. “If amply justified and grounded in empirical, real-world events, the concepts and theory generated to characterize a particular localized phenomenon [i.e., decision making regarding transfer of care] can be used to gain insight into other substantively different but theoretically comparable phenomena” (Eakin and Gladstone 2020, p2). 

Amended Text: No changes made to manuscript.

Comment 13: The authors did a good work in identifying the main factors implicated in their institution dynamics, presented their findings in an understandable form and acknowledged the limitations within the study itself, wich [sic] I believe can be recommended for publication as soon as the following minor points are solved: 

Author Response: We would like to thank Reviewer #3 for this positive comment and support for the publication of our work. 

Amended Text: N/A

Comment 14: A) Participants: it was chosen to have a purposive sample of staff involved in the studied behaviour. Please expand on this concept, explaining: 

Author Response: See response below for Comment Nos. 15-17. 

Amended Text: See relevant text changes below for Comment Nos. 15-17.

Comment 15: I) Why this type of sampling was optimal to the research goal, compared to others.

Author Response: Purposive sampling (also known as purposeful sampling) is a type of non-probability sampling widely used in qualitative research for the identification and selection of information-rich cases related to the phenomenon of interest (i.e., decision making regarding transfer of patient care. Purposive sampling method aligns with the research goal as it resulted in identifying and selecting individuals or groups of individuals that were especially knowledgeable about or experienced with the phenomenon of interest (i.e., those involved in making decisions regarding transfer of patient care from the ED to the MAU) while maximizing effective use of limited resources. In comparison, the ability to make accurate generalizations from random sampling is highly dependent on sample size. In qualitative research, however, “the sample sizes are typically so small that even random sampling would yield too little accuracy for meaningful generalizations” (Given 2008, p. 725). 

Amended Text: No changes made to manuscript.

Comment 16: II) If (and why) was any measure of bias reduction (I.E. sample-cherrypicking) taken, if any. 

Author Response: To clarify, the researchers did not ‘cherry-pick’ the sample. All senior staff directly involved in making decisions about transfer of care from ED to MAU (i.e., nurses, doctors, bed managers) were eligible to participate in the focus groups; identified the relevant Executive Director; and invited by email from the relevant Department Head to participate in the study. The sample included those eligible staff members who (voluntarily) attended at least one of the focus groups. We have amended the text to make this clearer to the reader. 

Amended Text: “Participants included a purposive sample of all senior staff involved in making or actioning decisions regarding hospital admission from the Emergency Medical Services (e.g., from the ED) to the Internal Medicine Service (e.g., to the MAU). The sample included those eligible staff members who attended at least one of the focus groups.”

Location of changes to manuscript: Section: Participants; Page: 10; lines: 170-174.

Author Response: In contrast to a one-to-one interview, in a focus group discussion, researchers adopt the role of a facilitator or moderator to facilitate or moderate a group discussion between participants, not between the researcher and the participants. The unit of measure is the focus group, not the individual staff member. 

Amended text: “Potential participants were identified by the Executive Directors of the two services and the sample included medical officers, bed managers, nurse shift coordinators, patient flow administrative officers, and nurse unit managers. Potential participants were invited to participate via an email invitation from the relevant Department head. All eligible participants were provided with a copy of the relevant Participant Information and Consent form(s) and informed consent was obtained prior to participation in the focus group. As part of the informed consent process, potential participants were advised that participation in the focus groups was voluntary and optional and that withdrawal from the study could occur at any time.” 

Location of changes to manuscript: Section: Recruitment and consent for focus groups; Page 10; lines: 175-184.

Author Response: We collected data across two sequential focus groups. In their analysis of 40 focus groups, Guest, Namey, McKenna (2017) have shown that more than 80% of all themes are discoverable within two to three focus groups.

Amended text: “The focus groups, conducted one after the other, were held during participants’ normal working hours, in agreement with the participants and their supervising line managers in order to minimize extra time burden for the participants.”

Location of change to manuscript: Section: Data collection; Page 10; Lines 187-190.

Comment 17: III) What was the proportion of the sample, in relation of the total staff population.

Author Response: As is the case with purposive sampling, the total population of staff is irrelevant. The focus group as a data collection method aims to obtain data from a purposely identified group of individuals (i.e., senior staff involved in making or actioning decisions regarding hospital admission from the ED to the MAU), rather than from a statistically representative sample of a broader population of ED and MAU staff. 

Amended Text: No changes made to manuscript.

Comment 18: B) The Authors correctly followed the COnsolidated criteria for REporting Qualitative studies (COREQ); however, not all COREQ Comments were reported in text. Some were N\\A, but two can be easily reported: 

Author Response: See responses below for Comment Nos. 19, 20. 

Amended Text: See relevant text changes for Comment Nos. 19, 20.

Comment 19: No.4) Gender 

Author Response: We have amended the text to include gender of the researchers. 

Amended Text: “The focus groups were attended in a conference room away from participants’ usual work areas and facilitated by a trained researcher (HC, a female social worker and academic), with a second trained researcher assisting and taking field notes (SRO, a female registered nurse and academic). At the start of each focus group, a third researcher (JdL, a male senior medical officer and clinician researcher) facilitated introductions and then left the room, leaving only participants present.”

Location of change to manuscript: Section: Data Collection; page: 10-11; lines: 190-197

Comment 20: No. 15) Presence of non-participants 

Author Response: We thank the reviewer for pointing out this missing detail and have amended the sentence in the revised manuscript. 

Amended Text: “At the start of each focus group, a third researcher (JdL, senior medical officer and clinician researcher), facilitated introductions and then left the room, leaving only participants present. HC and SO then introduced themselves, their roles in the research, and the purpose of the study at the start of each focus group.” 

Location of changes to text: Section: Data Collection; page: 11; lines: 193-197

Comment 21: Please provide such information in-text and update Additional File 2 accordingly. 

Author Response: Text has been amended as indicated. Additional File 2 (COREQ checklist has been incorporated into cover letter as Table 2.

Amended Text: See above and cover letter (Table 2)

JOURNAL REQUIREMENTS

Comment 22: (1) Please ensure that your manuscript meets PLOS ONE's style requirements, including those for file naming. The PLOS ONE style templates can be found at https://journals.plos.org/plosone/s/file?id=wjVg/PLOSOne_formatting_sample_main_body.pdf and https://journals.plos.org/plosone/s/file?id=ba62/PLOSOne_formatting_sample_title_authors_affiliations.pdf

Author Response: We have re-checked the manuscript against PLOS ONE’s style guide; and renamed files accordingly. 

Amended Text: Manuscript aligned to style templates throughout.

Comment 23: (2) Please include a copy of the interview guide used in the study, in both the original language and English, as Supporting Information, or include a citation if it has been published previously. 

Author Response: The interview guide was previously submitted as Additional File 1 in the original manuscript submission. We have re-uploaded again as Additional File 1 for your convenience. 

Amended Text: No changes made to manuscript.

Comment 24: (3) We note that you have indicated that data from this study are available upon request. PLOS only allows data to be available upon request if there are legal or ethical restrictions on sharing data publicly. For information on unacceptable data access restrictions, please see http://journals.plos.org/plosone/s/data-availability#loc-unacceptable-data-access-restrictions.

Author Response: While the study methodology can be replicated, the findings (i.e., data) are contextually based data from locally conducted focus groups. Reusing data in qualitative research raises epistemological, methodological, legal, and ethical issues (see Chauvette, Schick-Makaroff, Molzahn 2019). 

Firstly (epistemologically), qualitative data are inextricably connected to the context in which it was acquired and taking awary this contextual information will substantially inmpact the interpretation of the data, disconnecting it from its authentic meaning, possibly making it unusable (Chauvette, et al., 2019). 

Second (methodologically), some qualitative research is not conducive to secondary data analysis. The lived experience of participants is time-based and linked to the social, cultural, and political contexts of their lives. As such, the nuances and characteristics of the context-dependent knowledge may not be evident when data are reused (Chauvette, et al., 2019). 

Amended Text: These points have been addressed (as per our response) in the revised cover letter.

Comment 25: (3a) If there are ethical or legal restrictions on sharing a de-identified data set, please explain them in detail (e.g., data contain potentially identifying or sensitive patient information) and who has imposed them (e.g., an ethics committee). Please also provide contact information for a data access committee, ethics committee, or other institutional body to which data requests may be sent. 

Author Response: Third (legally and ethically), there are legal and ethical concerns as to whether the sharing and reuse of qualitative data can be done while still complying with legislation or the HREC. These concerns centre around informed consent and protection of privacy and confidentiality and there is question whether data can be reused without the involvement of the participants (Chauvette, et al., 2019).

The Participant Information and Consent Form (PICF) clause that specifies how the data from this qualitative research will be used is quite clear: “Your information will be used for the purpose of this research project and” possibly for the subsequent intervention development research study” (Participant Information Sheet/Consent Form TPCH v4_18Sep2017, p5). 

We hope that this detailed explanation accompanied by the specifics of the PICF that participants signed is sufficient to support our position on open sharing of the data set. 

We will address these points in our revised cover letter. 

Amended Text: These points have been addressed in the revised cover letter.

Comment 26: (3b) If there are no restrictions, please upload the minimal anonymized data set necessary to replicate your study findings as either Supporting Information files or to a stable, public repository and provide us with the relevant URLs, DOIs, or accession numbers. Please see http://www.bmj.com/content/340/bmj.c181.long for guidelines on how to de-identify and prepare clinical data for publication. For a list of acceptable repositories, please see http://journals.plos.org/plosone/s/data-availability#loc-recommended-repositories. 

Author Response: See explanatory response above. 

Amended Text: These points have been addressed in the revised cover letter.

Comment 27: (4) Please include captions for your Supporting Information files at the end of your manuscript, and update any in-text citations to match accordingly. Please see our Supporting Information guidelines for more information: http://journals.plos.org/plosone/s/supporting-information. 

Author Response: We first assumed that this comment related to our Additional Files. We have only submitted two additional files: (1) Additional File 1_Focus Group Guide, which is a text document and requires no caption; and (2) Additional File 2_The COREQ checklist, which we assumed to be for Editorial purpose only and not made publicly available. Nevertheless, we have added the following caption above the table but placed the COREQ Checklist at the end of the cover letter as Table 2.

If, on the other hand, this comment refers to the two Figure Files (i.e., Fig 1 and Fig 2), we did not put captions on these figures as per the instructions on the Author Guidelines (see image below).

Amended Text: No changes to Additional File_1. Caption added above table on Additional File_2 and table placed at the end of the cover letter.

Comment 28: Please review your reference list to ensure that it is complete and correct. 

Author Response: The reference list has been reviewed and deemed complete and correct.

Amended Text: Changes include replacing parentheses ‘()’ with brackets ‘[]’ around citation numbers.

REFERENCES

Chauvette A, Schick-Makaroff K, Molzahn AE. Open data in qualitative research. International Journal of Qualitative Methods 2019; first published January 22, 2019. doi:10.1177/1609406918823863

Eakin JM, Gladstone B. “Value-adding” Analysis: Doing more with qualitative data. International Journal of Qualitative Methods 2020; 19. First published on line 27 August. doi:10.1177/1609406920949333

Given L (ed). Random Sampling. In The SAGE Encyclopedia of Qualitative Research Methods. SAGE Publications, Inc. 2008; pp725-726. DOI: https://dx.doi.org/10.4135/9781412963909.n364

Guest G, Namey E, McKenna K. How many focus groups are enough? Building an evidence base for nonprobability sample sizes. Field Methods 2017; 29(1):3-22. https://doi.org/10.1177/1525822X16639015

Sofaer S. Qualitative methods: what are they and why use them?. Health Services Research 1999; 34(5 Pt 2), 1101–1118. https://www.ncbi.nlm.nih.gov/pmc/articles/PMC1089055/

Figure 1. Screen shot from Author Guidelines re figure captions [note: see response to reviewers file to view this figure; this is not a figure in the manuscript].

RESPONSE TO ANITA ESTES' email

Point 1) Thank you for including your ethics statement on the online submission form: "The study obtained ethical approval from The Prince Charles Hospital Human Research Ethics Committee (approval number: HREC/16/QPCH/365) and Queensland University of Technology (approval number: 1700000310). Written consent was obtained from participants prior to the focus groups and confirmed at the start of each focus group."

To help ensure that the wording of your manuscript is suitable for publication, would you please also add this statement at the beginning of the Methods section of your manuscript file.

Author Response: We had already included a statement about ethical approval in the Methods section, under the subheading Ethics. See Methods section, subsection Ethics, page 8, lines 147- 149. We have added the additional text and sentence requested under subsection “Recruitment and consent for focus groups” as it was the most logical place in the manuscript. 

Amendments/Changes: “All eligible participants were provided with a copy of the relevant Participant Information and Consent form(s) and informed consent was obtained prior to participation in the focus group. Written consent was obtained from participants prior to the focus groups and confirmed at the start of each focus group prior to participation in the focus group.”

Location of added text: Section: “Methods”; Subheading: “Recruitment and consent for focus groups”; page 10, lines 184-186.

Point 2) We note your current Data Availability Statement is: "Relevant data are within the manuscript in the form of participant quotes. Data cannot be shared publicly because of the nature of the qualitative data, which may not be generalizable across different settings, and the conditions of the ethics approval."

Additionally, we note that you have given three reasons (summarized below) why you will not share your data in your Cover Letter when re-submitting your revised manuscript: … 

Therefore, before we can proceed, please address the following prompts:

Point 2a.) Please upload the minimal data set as Supporting Information files or deposit them to a stable, public repository and provide us with the relevant URLs, DOIs, or accession numbers. For a list of recommended repositories, please see https://journals.plos.org/plosone/s/recommended-repositories.

Author Response: We randomly accessed 10 recently published papers by PloS One and could not see any reference to data availability or supporting files containing minimal data sets. We have selected option 2b. Please see response below.

Point 2b.) If there are ethical or legal restrictions on sharing a de-identified data set, please explain them in detail (e.g., data contain potentially sensitive information, data are owned by a third-party organization, etc.) and who has imposed them (e.g., an ethics committee. Please also provide non-author contact information* for a data access committee, ethics committee, or other institutional body to which data requests may be sent. 

Author Response: We provided an explanation to this query in the Response to Reviewers file and included it in our Revised cover letter. 

We subsequently consulted with the research governance office and our HREC (The Prince Charles Hospital Human Research Ethics Committee) about the request from PLoS One and our concerns that sharing data would violate our informed consent agreement with participants. We have received an email from the Research Governance Officer indicating that the consensus view following a departmental meeting resulted in the Governance Office “agree[ing] with our rationale regarding ethical considerations around publishing [our] qualitative data” and has responded, “In this circumstance, it was agreed that you [the authors] should include The Prince Charles HREC committee [TPCH HREC EC00168] as the institutional contact”. Further, we have an email from TPCH HREC “to confirm that TPCH HREC is happy to be nominated as the contact for data requests”. [Note we are happy to provide a copy of this email communication]. 

Amendments/Changes: We have revised our Data Availability Statement as follows:

Data Availability Statement

Requests in writing for data related to the study titled, Exploration of clinicians’ decision-making regarding transfer of patient care from the emergency department to a medical assessment unit: a qualitative study (HREC approval number HREC/16/QPCH/365) can be sent to:

The Prince Charles Hospital Human Research Ethics Committee (TPCH HREC EC00168)

Attention: Anne Carle, Executive Officer

Research, Ethics and Governance Unit, Building 14

The Prince Charles Hospital 

Rode Road

Chermside Qld 4032 Australia

Ph: (07) 3139 4500 

Research.TPCH@health.qld.gov.au

---

## [Decision Letter · Decision Letter 1]

20 Dec 2021

PONE-D-21-10148R1Exploration of clinicians’ decision-making regarding transfer of patient care from the emergency department to a medical assessment unit: a qualitative studyPLOS ONE

Dear Dr. Sonya Osborne,

Thank you for submitting your manuscript to PLOS ONE. After careful consideration, we feel that it has merit but does not fully meet PLOS ONE’s publication criteria as it currently stands. Therefore, we invite you to submit a revised version of the manuscript that addresses the points raised during the review process.

ACADEMIC EDITOR: Most of the comments have been addressed,however in order for the paper to be published, the Authors are asked to respond appropriately to the comments of Reviewer 3, in particular on sampling size and sampling strategy, using the suggestions given by Reviewer 3.The decision is justified on PLOS ONE’s publication criteria.   ======================

We look forward to receiving your revised manuscript.

Kind regards,

Filomena Pietrantonio

Academic Editor

PLOS ONE

Journal Requirements:

Additional Editor Comments (if provided):

Most of the comments have been addressed,however in order for the paper to be published, the Authors are asked to respond appropriately to the comments of Reviewer 3, in particular on sampling size and sampling strategy, using the suggestions given by Reviewer 3.

Reviewers' comments:

Reviewer's Responses to Questions

**Comments to the Author**

1. If the authors have adequately addressed your comments raised in a previous round of review and you feel that this manuscript is now acceptable for publication, you may indicate that here to bypass the “Comments to the Author” section, enter your conflict of interest statement in the “Confidential to Editor” section, and submit your "Accept" recommendation.

Reviewer #1: All comments have been addressed

Reviewer #3: (No Response)

2. Is the manuscript technically sound, and do the data support the conclusions?

Reviewer #1: Yes

Reviewer #3: Yes

3. Has the statistical analysis been performed appropriately and rigorously? 

Reviewer #1: Yes

Reviewer #3: N/A

4. Have the authors made all data underlying the findings in their manuscript fully available?

Reviewer #1: Yes

Reviewer #3: Yes

5. Is the manuscript presented in an intelligible fashion and written in standard English?

Reviewer #1: Yes

Reviewer #3: Yes

6. Review Comments to the Author

Reviewer #1: Dear, thank you for your revision. I appreciated the revised draft and I hope to read the published work soon

Reviewer #3: The authors addressed most of my previous comments, and I feel the quality of their work has improved. Some of the points previously raised, however, are yet to be improved: I am under the impression that the Authors interpreted some of my previos comments as an attempt of applying a quantitative methodology on their study, particularly regarding their sampling choices: I assure that this is not the case.

A) Sampling size.

The topic of optimal sample size in qualitative research is a debated one. As the authors wrote in their response, it is true that qualitative research is not seeking statistical significance, and that sample size has no major impact in the quality of the evidence provided. However, while it is true that even a single observation may lead to some good quality evidence, the claim that size has at least some significance in this type of research can be seen as well grounded, to the point that other authors even proposed suggestion regarding this precise matter. Cfr. (1) and (2).

The information on the sheer number of adherents to a qualitative research is an information a future reader may find useful to know, in the perspective of a future evaluation of the strength of the provided evidence regardless of its quality, and also, it furthers the goal of sustaining the most transparent behavioural practices in research.

Moreover, it is hard to conceive any legitimate reason not to disclose such an information, that albeit not vital for the research, is obviously known, available and easily for the Authors to provide.

Actually, the authors claim in their response that "The unit of measure is the focus group, not the individual staff member". Also in the paper, page 13 lines 241-242, they write: "Two focus groups were conducted back-to-back with a total of six and nine participants, respectively".

Since such information are already provided in-text, please amend the "Results" section in the abstract accordingly.

Also the same reasoning was behind the request of providing the information of the proportion of the size relatively to the number of eligible participants.

Again, the goal has never been the one of reaching a "statistical significance", but of providing the reader with the most comprehensive information of the research itself - even more so, since focus group composition and representativeness impact in the work generalizability - a term that, for our purposes, includes both credibility and transferability of the results. Cfr. (3), and (4).

B) Sampling strategy.

The authors did not provide any justification in choosing a purpositive sampling strategy, simply claiming that Purposive sampling is widely used and random sampling provide no actual benefits. Such propositions are well documented and adding this information in the text would raise its overall quality. Still, there are other sampling strategies the authors fail to address in their reasoning of why purpositive sampling was optimal to the research goal, most compared to other nonprobability sampling strategies. Cfr. (5).

All other comments have been adequately addressed by the Authors.

1: Malterud K, Siersma VD, Guassora AD. Sample Size in Qualitative Interview Studies: Guided by Information Power. Qual Health Res. 2016 Nov;26(13):1753-1760. doi: https://doi.org/10.1177/1049732315617444

2: Vasileiou, K., Barnett, J., Thorpe, S. et al. Characterising and justifying sample size sufficiency in interview-based studies: systematic analysis of qualitative health research over a 15-year period. BMC Med Res Methodol 18, 148 (2018). https://doi.org/10.1186/s12874-018-0594-7.

3: Andrew Parker & Jonathan Tritter (2006) Focus group method and methodology: current practice and recent debate, International Journal of Research & Method in Education, 29:1, 23-37, DOI: https://doi.org/10.1080/01406720500537304

4: Moon, Katie, et al. “A Guideline to Improve Qualitative Social Science Publishing in Ecology and Conservation Journals.” Ecology and Society, vol. 21, no. 3, Resilience Alliance Inc., 2016, http://www.jstor.org/stable/26269983

5: DeCarlo M, Scientific Inquiry in Social Work, August 7, 2018, Open Social Work Education, available at https://scientificinquiryinsocialwork.pressbooks.com/

7. PLOS authors have the option to publish the peer review history of their article (what does this mean?). If published, this will include your full peer review and any attached files.

Reviewer #1: No

Reviewer #3: **Yes: **Antonio Vinci

---

## [Author Response · Author response to Decision Letter 1]

4 Jan 2022

Response to Academic Editor and Reviewers

ACADEMIC EDITOR, COMMENT 1 - Most of the comments have been addressed… 

AUTHOR RESPONSE - We thank the Editor for this acknowledgement. 

CHANGE(S) TO MANUSCRIPT - No change to manuscript.

ACADEMIC EDITOR, COMMENT 2 - however, in order for the paper to be published, the Authors are asked to respond appropriately to the comments of Reviewer 3, in particular on sampling size and sampling strategy, using the suggestions given by Reviewer 3.

AUTHOR RESPONSE - We believe we have responded appropriately to Reviewer #3’s comments to date and that our responses on the most recent comments (see below) are equally appropriate, particularly in relation to sample size and sampling strategy, using suggestions by Reviewer #3, where deemed appropriate.

CHANGE(S) TO MANUSCRIPT - Response to comments and changes to manuscript, where applicable, are identified below.

ACADEMIC EDITOR, COMMENT 3 - The decision is justified on PLOS ONE’s publication criteria. 

AUTHOR RESPONSE - We believe our manuscript is aligned with the Submission Guidelines in general, but specifically, in relation to the special section called Qualitative Research as outlined below:

Qualitative research

“Qualitative research studies use non-quantitative methods to address a defined research question that may not be accessible by quantitative methods, such as people's interpretations, experiences, and perspectives…” 

A point we made and support with evidence from the literature in our previous responses.

“…The analysis methods are explicit, systematic, and reproducible, but the results do not involve numerical values or use statistics…”

We believe we have sufficiently described the explicit and systematic approach of our methodology and methods and presented results in text, as is conventional when reporting qualitative research results.

Further, as per the guidelines in particular relation to qualitative research studies, we “reported [our study] in accordance to the Consolidated criteria for reporting qualitative research (COREQ) checklist…” 

(Source of cited text: PLOS ONE: accelerating the publication of peer-reviewed science)

CHANGE(S) TO MANUSCRIPT - No change to manuscript.

REVIEWER #1, COMMENT 1 - Dear, thank you for your revision. I appreciated the revised draft and I hope to read the published work soon.

AUTHOR RESPONSE - We thank Reviewer #1 for the time taken to review our manuscript and for the constructive feedback to date. We, too, hope to read the published work soon.

CHANGE(S) TO MANUSCRIPT - No change to manuscript.

REVIEWER #3, COMMENT 1 - The authors addressed most of my previous comments, and I feel the quality of their work has improved. Some of the points previously raised, however, are yet to be improved: I am under the impression that the Authors interpreted some of my previous [sic] comments as an attempt of applying a quantitative methodology on their study, particularly regarding their sampling choices: I assure that this is not the case.

AUTHOR RESPONSE - We would like to remind Reviewer # 3 of his initial comments on the original manuscript – “The methodology is not particularly strong however, due to the design being a qualitative research conducted on a sample from a single institution.” 

We feel there is no other way to interpret this comment accept on face value, that is, as a denigration and/or deprecation of qualitative research methodology. Consequently, this comment set the tone for all further comments by Reviewer #3, including the comments about sampling choices.

CHANGE(S) TO MANUSCRIPT - No change to manuscript.

REVIEWER #3, COMMENT 2 - A) Sampling size.

The topic of optimal sample size in qualitative research is a debated one.

AUTHOR RESPONSE - We agree that the topic of optimal sample size in qualitative research is a debated one. Given that we agree on this point, we feel continuing the debate on this topic between Reviewer #3 and our research team is futile at this point. We recommend that we ‘agree to disagree’ on the approach to sample size and sampling in qualitative research.

We have made changes in regards to sample size throughout the manuscript in response to specific comments as outlined below.

REVIEWER #3, COMMENT 3 - As the authors wrote in their response, it is true that qualitative research is not seeking statistical significance, and that sample size has no major impact in the quality of the evidence provided. However, while it is true that even a single observation may lead to some good quality evidence, the claim that size has at least some significance in this type of research can be seen as well grounded, to the point that other authors even proposed suggestion regarding this precise matter. Cfr. [Malterud et al 2016] and [2: Vasileiou et al 2018]. 

AUTHOR RESPONSE - We concede that more information on choices made regarding sample size would benefit the reader and aid in transparency.

CHANGE(S) TO MANUSCRIPT - We have made changes in regards to sample size throughout the manuscript in response to specific comments as outlined below.

REVIEWER #3, COMMENT 4 - The information on the sheer number of adherents to a qualitative research is an information a future reader may find useful to know, in the perspective of a future evaluation of the strength of the provided evidence regardless of its quality, and also, it furthers the goal of sustaining the most transparent behavioural practices in research.

AUTHOR RESPONSE - While the intention of our manuscript is not to produce a methodological paper, we agree with this insightful comment and acknowledge that any explanations or rationale for choices made in research would be of benefit to the reader.

CHANGE(S) TO MANUSCRIPT - We have added further explanations on qualitative research processes used in the study to make this transparent and to benefit the reader.

Topic: Reflexivity

Subheading: Data analysis

Page: 13

Lines 239-243

Topic: Triangulation of data from different sources

Subheading: Discussion

Page: 33

Lines 663-665, 668-669

Topic: Triangulation of data from different sources

Subheading: Discussion

Page: 35

Lines 714-718

Changes highlighted in manuscript.

REVIEWER #3, COMMENT 5 - Moreover, it is hard to conceive any legitimate reason not to disclose such an information, that albeit not vital for the research, is obviously known, available and easily for the Authors to provide.

Actually, the authors claim in their response that "The unit of measure is the focus group, not the individual staff member". Also in the paper, page 13 lines 241-242, they write: "Two focus groups were conducted back-to-back with a total of six and nine participants, respectively".

Since such information are already provided in-text, please amend the "Results" section in the abstract accordingly.

AUTHOR RESPONSE - We have amended the results section of the abstract as suggested.

In order to keep to the abstract length requirement of 300 words, we have re-crafted the entire abstract to accommodate the additional text. Abstract word count is 286.

CHANGE(S) TO MANUSCRIPT - Location in manuscript:

Subheading: Abstract

Pages: 3-4

Lines: 25-47

Changes highlighted in manuscript.

REVIEWER #3, COMMENT 6 - Also the same reasoning was behind the request of providing the information of the proportion of the size relatively to the number of eligible participants. Again, the goal has never been the one of reaching a "statistical significance", but of providing the reader with the most comprehensive information of the research itself - even more so, since focus group composition and representativeness impact in the work generalizability - a term that, for our purposes, includes both credibility and transferability of the results. Cfr. [3: Parker and Tritter 2006] and [4: Moon et al 2016].

AUTHOR RESPONSE - We agree that providing the reader with the most comprehensive information of the research itself would enhance the manuscript. We have, thus, added more detail about our methods, findings and limitations to aid transparency and transferability of our methods.

CHANGE(S) TO MANUSCRIPT - In the following section we have provided more detail on the justification for our methodological choices regarding sample size.

Location in Manuscript:

Subheading: Findings

Pages 13-14

Lines: 257-264

Location in Manuscript:

Subheading: Limitations

Pages: 38-39

Lines: 786-809

Changes highlighted in manuscript. 

REVIEWER #3, COMMENT 7 – [re] B) Sampling strategy.

The authors did not provide any justification in choosing a purpositive [sic] sampling strategy, simply claiming that Purposive sampling is widely used and random sampling provide no actual benefits. 

REVIEWER #3, COMMENT 8 - Such propositions are well documented and adding this information in the text would raise its overall quality. 

REVIEWER #3, COMMENT 9 - Still, there are other sampling strategies the authors fail to address in their reasoning of why purpositive sampling was optimal to the research goal, most compared to other nonprobability sampling strategies. Cfr. (5: DeCarlo 2018).

AUTHOR RESPONSE - We are satisfied with our previous responses to Reviewer #3 on the topic of sampling and sample size in qualitative research. We agree that adding the information in the text would enhance the manuscript. We have, thus, incorporated our previous responses into the manuscript to enhance transparency and transferability of our methods.

CHANGE(S) TO MANUSCRIPT – 

Location in Manuscript:

Subheading: Participants

Page: 10

Lines: 173-183

Changes highlighted in manuscript. 

REVIEWER #3, COMMENT 10 - All other comments have been adequately addressed by the Authors

AUTHOR RESPONSE - We thank Reviewer #3 for the time taken to review our original manuscript and subsequent revisions. 

CHANGE(S) TO MANUSCRIPT – No change to manuscript.

---

## [Decision Letter · Decision Letter 2]

17 Jan 2022

Exploration of clinicians’ decision-making regarding transfer of patient care from the emergency department to a medical assessment unit: a qualitative study

PONE-D-21-10148R2

Dear Dr.Sonya Osborne,

We’re pleased to inform you that your manuscript has been judged scientifically suitable for publication and will be formally accepted for publication once it meets all outstanding technical requirements.

Kind regards,

Filomena Pietrantonio

Academic Editor

PLOS ONE

Additional Editor Comments (optional):

All comments have been addressed and the paper is now suitable for publication. The second reviewer had already communicated that the paper was suitable for publication

Reviewers' comments:

Reviewer's Responses to Questions

**Comments to the Author**

1. If the authors have adequately addressed your comments raised in a previous round of review and you feel that this manuscript is now acceptable for publication, you may indicate that here to bypass the “Comments to the Author” section, enter your conflict of interest statement in the “Confidential to Editor” section, and submit your "Accept" recommendation.

Reviewer #3: All comments have been addressed

2. Is the manuscript technically sound, and do the data support the conclusions?

Reviewer #3: Yes

3. Has the statistical analysis been performed appropriately and rigorously? 

Reviewer #3: Yes

4. Have the authors made all data underlying the findings in their manuscript fully available?

Reviewer #3: Yes

5. Is the manuscript presented in an intelligible fashion and written in standard English?

Reviewer #3: Yes

6. Review Comments to the Author

Reviewer #3: The Authors assessed all comments in a satisfactory way and I believe the paper can be accepted for pubblication.

7. PLOS authors have the option to publish the peer review history of their article (what does this mean?). If published, this will include your full peer review and any attached files.

Reviewer #3: **Yes: **Antonio Vinci

---

## [Editor Report · Acceptance letter]

24 Jan 2022

PONE-D-21-10148R2 

Exploration of clinicians’ decision-making regarding transfer of patient care from the emergency department to a medical assessment unit: a qualitative study 

Dear Dr. Osborne:

I'm pleased to inform you that your manuscript has been deemed suitable for publication in PLOS ONE. Congratulations! Your manuscript is now with our production department. 

Kind regards, 

on behalf of

Dr. Filomena Pietrantonio 

Academic Editor

PLOS ONE